# Adaptive social contact rates induce complex dynamics during epidemics

Ronan F. Arthur[1], James H. Jones[2], Matthew H. Bonds[3], Yoav Ram[4,5,6], Marcus W. Feldman[7]*

1 School of Medicine, Stanford University, Stanford, California, United States of America, 2 Department of Earth Systems Science, Stanford University, Stanford, California, United States of America, 3 Department of Global Health and Social Medicine, Harvard Medical School, Cambridge, Massachusetts, United States of America, 4 School of Computer Science, Interdisciplinary Center Herzliya, Herzliya, Israel, 5 School of Zoology, Faculty of Life Sciences, Tel Aviv University, Tel Aviv, Israel, 6 Sagol School of Neurosciences, Tel Aviv University, Tel Aviv, Israel, 7 Department of Biology, Stanford University, Stanford, California, United States of America

* mfeldman@stanford.edu

**Data Availability Statement:** All relevant data are within the manuscript and its Supporting information files.

**Funding:** This research was supported in part by the Morrison Institute for Population and Research

## Abstract

Epidemics may pose a significant dilemma for governments and individuals. The personal or public health consequences of inaction may be catastrophic; but the economic consequences of drastic response may likewise be catastrophic. In the face of these trade-offs, governments and individuals must therefore strike a balance between the economic and personal health costs of reducing social contacts and the public health costs of neglecting to do so. As risk of infection increases, potentially infectious contact between people is deliberately reduced either individually or by decree. This must be balanced against the social and economic costs of having fewer people in contact, and therefore active in the labor force or enrolled in school. Although the importance of adaptive social contact on epidemic outcomes has become increasingly recognized, the most important properties of coupled human-natural epidemic systems are still not well understood. We develop a theoretical model for adaptive, optimal control of the effective social contact rate using traditional epidemic modeling tools and a utility function with delayed information. This utility function trades off the population-wide contact rate with the expected cost and risk of increasing infections. Our analytical and computational analysis of this simple discrete-time deterministic strategic model reveals the existence of an endemic equilibrium, oscillatory dynamics around this equilibrium under some parametric conditions, and complex dynamic regimes that shift under small parameter perturbations. These results support the supposition that infectious disease dynamics under adaptive behavior change may have an indifference point, may produce oscillatory dynamics without other forcing, and constitute complex adaptive systems with associated dynamics. Implications for any epidemic in which adaptive behavior influences infectious disease dynamics include an expectation of fluctuations, for a considerable time, around a quasi-equilibrium that balances public health and economic priorities, that shows multiple peaks and surges in some scenarios, and that implies a high degree of uncertainty in mathematical projections.

Studies (morrisoninstitute.stanford.edu) (MWF), by Israel Science Foundation grants 552/19 and 3811/19 (isf.org.il) (YR), and by a Graduate Research Fellowship from the National Science Foundation #2015160091 (nsf.gov) (RFA). The funders had no role in study design, data collection and analysis, decision to publish, or preparation of the manuscript.

**Competing interests:** The authors have declared that no competing interests exist.

## Author summary

Epidemic response in the form of social contact reduction, such as has been utilized during the ongoing COVID-19 pandemic, presents inherent tradeoffs between the economic costs of reducing social contacts and the public health costs of neglecting to do so. Such tradeoffs introduce an interactive, iterative mechanism that adds complexity to an infectious disease system. Consequently, infectious disease modeling typically has not included dynamic behavior change that must address such a tradeoff. Here, we develop a theoretical strategic model that introduces lost or gained economic and public health utility through the adjustment of social contact rates with delayed information. This model produces an equilibrium, a point of indifference where the tradeoff is neutral, and at which a disease will be endemic for a long period of time. Under small perturbations, this model exhibits complex dynamic regimes, including oscillatory behavior, runaway exponential growth, and eradication. These dynamics suggest that for epidemic responses that rely on social contact reduction, secondary waves and surges with accompanied business and school re-closures and shutdowns may be expected, and that accurate projection under such circumstances is unlikely.

## Introduction

Adapting to a changing landscape of risk during an infectious disease epidemic may pose a significant dilemma for a susceptible individual or for a governing body responsible for the health of susceptible individuals. On the one hand, changing behavior (e.g. through social distancing) can reduce the reproduction number ($R_0$) of an epidemic and save many from death or morbidity [1, 2]. On the other hand, behavior change can reduce an individual's ability to make a living or, for a group of people, can hamper or cause a recession in the economy through decreased production, sales, and investment and increased unemployment, inflation, and debt [3]. This dilemma introduces a behavior change tradeoff for the decision-maker, a balancing act between epidemiological interests and economic interests.

There is growing interest in the role of behavior in infectious disease dynamics (see Funk et al., 2010 [4] for a general review). Behavior relevant to epidemic outcomes is known to change in response to perceived risk during epidemics (e.g. measles-mumps-rubella (MMR) vaccination choices [5], condom purchases in HIV-affected communities [6], and social distancing in influenza outbreaks [7] and during the ongoing COVID-19 pandemic [8]). Although behavior is difficult to measure, quantify, and predict [9], modelers have adopted a variety of strategies to investigate its role in epidemic outcomes. These strategies include agent-based modeling [10], network structures that model behavior as a social contagion process [11] or that replace central nodes when sick [12], and game theoretic descriptions of rational choice under changing incentives, as in the case of vaccination [7, 13, 14]. A common approach to incorporating behavior into epidemic models is to track co-evolving dynamics of behavior and infection [11, 15–17].

In epidemic response policy, it is typical to think of behavior change as an exogenously-induced intervention without considering associated incentives for the individual or the collective. Due to the interactive relationship between behavior and epidemic dynamics, adaptive behavior should instead be thought of as endogenous to an infectious disease system because it

is, in part, a consequence of the prevalence of the disease, which in turn responds to changes in behavior [9, 18]. An epidemic system with adaptive behavior responds to the conditions it itself creates, and is thus a complex, adaptive system [19], subject to the properties and tendencies of such systems.

The interaction between behavioral incentives and epidemic dynamics introduces a negative feedback into the epidemic system. In an important early expansion of Kermack and McKendrick's seminal Susceptible-Infectious-Removed (SIR) model [20], Capasso and Serio built a self-iterative epidemic model by making the transmission parameter ($\beta$) a negative function of the number of infected because "in the presence of a very large number of infectives the population may tend to reduce the number of contacts per unit time." [21] A negative feedback such as this may lead to an endemic equilibrium [22]. This happens because, at low levels of prevalence, the cost of behavior change to avoid disease relative to the risk of infection may not be justified, even though the collective, public benefit in the long-term may be greater. Conversely, as prevalence increases, the probability of infection also increases, thus increasing incentives to adopt protective behavior [13]. If responses are based on outdated information, a negative feedback between prevalence and social contact can produce sustained oscillations in time-series data [23].

Such periodicity (i.e. multi-peak dynamics) has long been documented empirically in epidemiology [24, 25]. Periodicity can be driven by seasonal contact rate changes (e.g. when children are in school) [26], seasonality in the climate or ecology [27], sexual and social behavior change [23, 28], and host immunity cycling through new births of susceptibles or a decay of immunity over time. Some papers in nonlinear dynamics have studied delay differential equations in the context of epidemic dynamics and found periodic solutions as well [29]. Although it is atypical to include delay in modeling, delay is an important feature of epidemics. Delays of information acquisition, behavioral response, scientific investigation, and those inherent in natural biological processes can affect epidemic outcomes. In the ongoing COVID-19 pandemic, for example, there have been delays in the international recognition of the outbreak [30], delays in the identification of the virus, delays in the acquisition of reliable information on suspected and confirmed cases [31], and delays in the development and deployment of competent diagnostics [32].

Although infectious disease modelers have begun to incorporate adaptive behavior into their models, few studies in the literature capture the competing economic and public health incentives that drive delayed behavioral responses in both individual and group settings during epidemics [33, 34]. Here we develop a theoretical model using both discrete and continuous time and both SIR and SIS compartmental epidemic structures. The model, which is designed to be strategic rather than tactical (sensu Holling [35]), is adjusted on the principle of endogenous behavior change through an adaptive social-contact rate that can be thought of as either individually motivated or institutionally imposed. We introduce a novel utility function that motivates the population's effective contact rate at a particular time period. This utility function is based on information about the epidemic size that may not be current. This leads to a time delay in the contact function that increases the complexity of the population dynamics of the infection. Results from the discrete-time model show that the system approaches an equilibrium in many cases, although small parameter perturbations can lead the dynamics to enter qualitatively distinct regimes. The analogous continuous-time model retains periodicities for some sets of parameters, but numerical investigation shows that the continuous time version is much better behaved than the discrete-time model. This behavior is similar to that in models of ecological population dynamics, and a useful mathematical parallel can be drawn between these systems.

## Model specifications

### SIS

To represent endogenous behavior change, we start with the classical discrete-time suscepti-ble-infected-susceptible (SIS) model [20], which, when incidence is relatively small compared to the total population [36, 37], can be written in terms of the recursions

$$S_{t+1} = S_t - b \ S_t I_t + \gamma I_t \tag{1}$$

$$I_{t+1} = I_t + b \ S_t I_t - \gamma I_t \tag{2}$$

$$S_t + I_t = N_t, \tag{3}$$

where at time t, $S_t$ represents the number of susceptible individuals, $I_t$ the infected individuals, and $N_t$ the number of individuals that make up the population, which is assumed fixed in a closed population. We can therefore write N for the constant population size. Here $\gamma$, with $0 < \gamma < 1$, is the rate of removal from $I$ to $S$ due to recovery. This model in its simplest form assumes random mixing, where the parameter $b$ represents a composite of the average contact rate and the disease-specific transmissibility given a contact event. In order to introduce human behavior, we substitute for $b$ a time-dependent $b_t$, which is a function of both $b_0$, the probability that disease transmission takes place on contact, and a dynamic social contact rate $c_t$ whose optimal value, $c_t^*$, is the number of contacts per unit time that maximize utility for the individual. $c_t^*$ is determined at each time $t$ as in economic epidemiological models [34], namely

$$b_t = b_0 \ c_t^*, \tag{4}$$

where $c_t^*$ represents the optimal contact rate, defined as the number of contacts per unit time that maximize utility for the individual. Here, $c_t^*$ is a function of the number of infected in the population according to the perceived risks and benefits of social contacts, which we model as a utility function. We assume that there is a constant utility independent of contact, a utility loss associated with infection, and a utility derived from the choice of number of daily contacts with a penalty for deviating from the choice of contacts which would yield the most utility.

This utility function is assumed to take the form

$$U(c) = \alpha_0 - \alpha_1 (c - \hat{c})^2 - \alpha_2 \left\{ 1 - \left[ 1 - \left( \frac{I_{t-\Delta}}{N} \right) b_0 \right]^c \right\}. \tag{5}$$

Here $U$ represents utility for an individual at time t given a particular number of contacts per unit time $c$, $\alpha_0$ is a constant that represents maximum potential utility achieved at a target contact rate $\hat{c}$. The second term, $-\alpha_1 (c - \hat{c})^2$, is a concave function that represents the penalty for deviating from $\hat{c}$. The third term, $\alpha_2 \left\{ 1 - \left[ 1 - \left( \frac{I_{t-\Delta}}{N} \right) b_0 \right]^c \right\}$, is the cost of infection (i.e. mor-bidity), $\alpha_2$, multiplied by the probability of infection over the course of the time unit. The time-delay $\Delta$ represents the delay in information acquisition and the speed of response to that information. We note that $\left( 1 - \frac{I}{N} b_0 \right)^c$ can be approximated by

$$\left[ 1 - \left( \frac{I}{N} \right) b_0 \right]^c \approx 1 - c \left( \frac{I}{N} \right) b_0, \tag{6}$$

when $\frac{I}{N} b_0$ is small and $c \frac{I}{N} b_0 \ll 1$. We thus assume $\frac{I}{N} (b_0)$ is small, that is, prevalence is low, and approximate $U(c)$ in Eq 5 using Eq 6. Eq 5 assumes a strictly negative relationship between number of infecteds and contact.

We assume an individual or government will balance the cost of infection, the probability of infection, and the cost of deviating from the target contact rate $\hat{c}$ to select an optimal contact rate $c_t^*$, namely the number of contacts, which takes into account the risk of infection and the penalty for deviating from the target contact rate. This captures the idea that individuals trade off how many people they want to interact with versus their risk of becoming infected, or that authorities want to reopen the economy during a pandemic and have to trade off morbidity and mortality from increasing infections with the need to allow additional social contacts to help the economy restart. This optimal contact rate can be calculated by finding the maximum of $U$ with respect to $c$ from Eq 5 with substitution from Eq 6, namely

$$U(c) = \alpha_0 - \alpha_1(c - \hat{c})^2 - \alpha_2 c\left(\frac{I_{t-\Delta}}{N}\right)b_0. \tag{7}$$

Differentiating, we have

$$\frac{dU(c)}{dc} = -2\alpha_1(c - \hat{c}) - \alpha_2 b_0 \frac{I_{t-\Delta}}{N}, \tag{8}$$

which vanishes at the optimal contact rate, $c^*$, which we write as $c_t^*$ to show its dependence on time. Then

$$c_t^* = \hat{c} - \frac{\alpha_2}{2\alpha_1}b_0 \frac{I_{t-\Delta}}{N}, \tag{9}$$

which we assume to be positive. Therefore, total utility will decrease as $I_t$ increases and $c_t^*$ also decreases. Utility is maximized at each time step, rather than over the course of lifetime expectations. In addition, Eq 9 assumes a strictly negative relationship between number of infecteds at time $t - \Delta$ and $c_t^*$. While behavior at high degrees of prevalence has been shown to be non-linear and fatalistic [38, 39], in this model, prevalence (i.e., $\frac{b_0 I_t}{N}$) is assumed to be small, consistent with Eq 6.

We introduce the new parameter $\alpha = \frac{\alpha_2}{2\alpha_1}b_0$, so that

$$c_t^* = \hat{c} - \alpha \frac{I_{t-\Delta}}{N}. \tag{10}$$

We can now rewrite the recursion from Eq 2, using Eq 4 and replacing $c_t$ with $c_t^*$ as defined by Eq 10, as

$$I_{t+1} = I_t^2\left(\frac{b_0\alpha}{N}I_{t-\Delta} - b_0\hat{c}\right) + I_t(b_0 N\hat{c} - \alpha b_0 I_{t-\Delta} + 1 - \gamma) = f(I_t, I_{t-\Delta}). \tag{11}$$

When $\Delta = 0$ and there is no time delay, $f(\cdot)$ is a cubic polynomial, given by

$$f(I_t) = \frac{b_0\alpha}{N}I_t^3 - b_0(\hat{c} + \alpha)I_t^2 + (Nb_0\hat{c} + 1 - \gamma)I_t. \tag{12}$$

## SIR

For the susceptible-infected-removed (SIR) version of the model, we include the removed category and write the (discrete-time) recursion system as

$$S_{t+1} = S_t - b_t S_t I_t \tag{13}$$

$$I_{t+1} = I_t + b_t S_t I_t - \gamma I_t \tag{14}$$

$$R_{t+1} = R_t + \gamma I_t, \tag{15}$$

where $R_t = N - I_t - S_t$, $b_t = b_0 c_t^*$ with $b_0$ the baseline contact rate and $c_t^*$ specified by Eq 10. With $b_t = b$, say, and not changing over time, Eqs 13–15 form the discrete-time version of the classical Kermack-McKendrick SIR model [20]. The inclusion of the removed category entails that $\tilde{I} = 0$ is the only equilibrium of the system Eqs 13–15; unlike the SIS model, there is no equilibrium with infecteds present. In general, since $c_t^*$ includes the delay $\Delta$, the dynamic approach to $\tilde{I} = 0$ is expected to be quite complex. Numerical analysis of this SIR model shows strong similarity between the SIS and SIR models for several hundred time steps before the SIR model converges to $\tilde{I} = 0$. In the section "Numerical Iteration and Continuous-Time Analog" we compare the numerical iteration of the SIS (Eq 11) and SIR (Eqs 13–15) and integration of the continuous-time (differential equation) versions of the SIS and SIR models.

## Analytical results

### Equilibria

To determine the dynamic trajectories of (11) without time delay, we first solve for the fixed point(s) of the recursion (11) (i.e., value or values of $I$ such that $f(I_{t+1}) = I_t = I_{t-\Delta}$). That is, we solve

$$I = \frac{b_0 \alpha}{N} I^3 - b_0(\hat{c} + \alpha)I^2 + (N b_0 \hat{c} + 1 - \gamma)I. \tag{16}$$

From Eq 16, it is clear that $I = 0$ is an equilibrium as no new infections can occur in the next time-step if none exist in the current one. This is the disease-free equilibrium denoted by $\tilde{I}$. Other equilibria are the solutions of

$$\frac{b_0 \alpha}{N} I^2 - b_0(\hat{c} + \alpha)I + N b_0 \hat{c} - \gamma = 0, \tag{17}$$

namely

$$\frac{\alpha + \hat{c} \pm \sqrt{(\alpha - \hat{c})^2 + \frac{4\alpha\gamma}{N b_0}}}{2\alpha/N}. \tag{18}$$

We label the solution with the + sign $I^*$ and the one with the − sign $\hat{I}$. $I^* > 0$ but $I^* \leq N$ if $4\alpha\gamma/N b_0 \leq 0$, which is impossible under our assumptions that $\alpha$ and $\gamma$ are positive. Hence $I^*$ is not feasible. Further, under these same conditions, $\hat{I} \leq N$, and $\hat{I} > 0$ if

$$N\hat{c}b_0 > \gamma. \tag{19}$$

It is important to note that under these conditions $\hat{I}$ is an equilibrium of the recursion (11) for any $\Delta \geq 0$. Recall that for the SIR version of this model the only equilibrium is $\tilde{I} = 0$.

## Stability of the equilibria

Assessing global asymptotic stability in epidemic models is an important task of mathematical epidemiology [40, 41]. The three equilibria of the SIS recursion (11) are qualitatively different. $\tilde{I} = 0$ corresponds to a disease-free population; $I^*$ is greater than $N$ and is therefore not feasible; $\hat{I}$ is the only positive feasible equilibrium if $\hat{c}b_0 > \gamma/N$ (this is equivalent to $R_0 > 1$, where $R_0 = N\hat{c}b_0 + 1 - \gamma$) and is, therefore, the most interesting for the asymptotic stability behavior of the epidemic. Mathematical stability analysis of recursion (11) is complicated because of the delay term $\Delta$. However, from (11), if $N\hat{c}b_0 > \gamma$, the disease-free equilibrium $\tilde{I} = 0$ is locally unstable, and in this case $\hat{I}$ is indeed feasible.

Local stability of $\hat{I}$ in (18) is discussed in detail in S1 Appendix. First, in the absence of delay (i.e., $\Delta = 0$), $\hat{I}$ is locally stable if $\left| \frac{d}{dI} f(I) \right|_{I=\hat{I}} < 1$, and the condition for this to hold when $\hat{I}$ is legitimate is

$$b_0 \hat{I} \sqrt{(\alpha - \hat{c})^2 + \frac{4\alpha\gamma}{Nb_0}} < 2. \tag{20}$$

If inequalities (20) and $N\hat{c}b_0 > \gamma$ hold, then $\hat{I}$ is locally stable. However, even if both of these inequalities hold, the number of infecteds may not converge to $\hat{I}$. It is well known that iterations of discrete-time recursive relations, of which (12) is an example (i.e., with $\Delta = 0$), may produce cycles or chaos depending on the parameters and the starting frequency $I_0$ of infecteds.

## Numerical iteration and continuous-time analog

We begin with numerical analysis of the discrete-time SIS recursion (11), which includes the delay parameter $\Delta$. Local stability properties of the equilibrium state $\hat{I}$, with $0 < \hat{I} < N$, are shown in the Appendix under the assumption $N\hat{c}b_0 > \gamma$, which also entails that the disease-free equilibrium $\tilde{I} = 0$ is locally unstable. In the recursion (11), the number of infecteds at time $t$ will not, in general, be integers, but can be interpreted as the expected number of infected in the population. Further, the dynamics of $I_t$ under such a recursion can be very sensitive to the starting condition $I_0$, the size of the time delay $\Delta$, and the parameters: $N, b_0, \gamma, \hat{c}$, and $\alpha$. The local stability of $\hat{I}$, namely whether $I_t$ converges to $\hat{I}$ from a starting number of infecteds close to $\hat{I}$, may tell you little about the actual trajectory of $I_t$ from other starting conditions.

Table 1 reports an array of dynamic trajectories without delay ($\Delta = 0$) for some choices of parameters. In seven cases, $I_0 = 1$, and in two cases the numerical iteration of Eq 12 was initiated with $I_0 \neq 1$. The first three rows show three sets of parameters for which the equilibrium values of $\hat{I}$ are very similar but the trajectories of $I_t$ are different: a two-point cycle, a four-point cycle, and apparently chaotic cycling above and below $\hat{I}$. In all of these cases, $df(I)/dI|_{I=\hat{I}} < -1$. Clearly the dynamics are sensitive to the target contact rate $\hat{c}$ in these cases. The fourth and eighth rows show that $I_t$ becomes unbounded (tends to $+\infty$) from $I_0 = 1$, but a two-point cycle is approached if $I_0$ is close enough to $\hat{I} : df(I)/dI|_{I=\hat{I}} < -1$ in these cases. For the parameters in the ninth row, if $I_0$ is close enough to $\hat{I}$ there is damped oscillation into $\hat{I}$: here $-1 < df(I)/dI|_{I=\hat{I}} < 0$. In the case marked $^*$, $\hat{I}$ locally stable and with a large enough initial number of infecteds, there is damped oscillatory convergence to $\hat{I}$. In the case marked $^{**}$, with $I_0 = 1$ the number of infecteds becomes unbounded, but in this case, $\hat{I}$ is locally unstable ($df(I)/dI|_{I=\hat{I}} < -1$), and starting from $I_0$ close to $\hat{I}$ a stable two-point cycle is approached.

**Table 1. Some results for dynamics of infection with Δ = 0.**

| Parameters | | | | | Equilibrium | |
|---|---|---|---|---|---|---|
| $N$ | $b_0$ | $\gamma$ | $\hat{c}$ | $\alpha$ | $\hat{I}$ | Dynamics |
| 250 | 0.1 | 0.1 | 0.2 | 0.1 | 240.371 | $I_0 = 1$: two-point cycle 110.436, 339.564 |
| 250 | 0.1 | 0.1 | 0.205 | 0.1 | 240.799 | $I_0 = 1$: four-point cycle above and below $\hat{I}$ |
| 250 | 0.1 | 0.1 | 0.209 | 0.1 | 241.115 | $I_0 = 1$: apparent chaos around $\hat{I}$ |
| 250 | 0.5 | 0.1 | 0.1 | 0.1 | 227.639 | $I_0 = 1$: becomes unbounded. $I_0 = 226$: converges to two-point cycle. |
| 250 | 0.115 | 0.1 | 0.1 | 0.1 | 203.375 | $I_0 = 1$: overshoots $\hat{I}$, then decreases to $\hat{I}$ |
| 350 | 0.1 | 0.1 | 0.1 | 0.1 | 290.839 | $I_0 = 1$: overshoots $\hat{I}$, then decreases to $\hat{I}$ |
| 1,000 | 0.1 | 0.1 | 0.1 | 0.1 | 900.000 | $I_0 = 1$: damped oscillation to $\hat{I}$ |
| 1,100 | 0.1 | 0.1 | 0.1 | 0.1 | 995.119 | $I_0 = 1$: $I_t$ becomes unbounded $I_0 = 990$: damped oscillation to $\hat{I}$ |
| 10,000 | 0.05 | 0.08 | 0.0015 | 0.375 | 35.718 | $I_0 = 1$: monotone convergence to $\hat{I}$ |

S5 Fig is a bifurcation diagram for recursion (11) with Δ = 0 and the other parameters from the first three lines of Table 1. As $\hat{c}$ increases, first there is convergence to $\hat{I}$, then period doubling to chaos and finally passage to negative infinity.

Stability analysis of the SIS model is more complicated when Δ ≠ 0, and in S1 Appendix we outline the procedure for local analysis of the recursion (11) near $\hat{I}$. Local stability is sensitive to the delay time Δ as can be seen from the numerical iteration of (11) for the specific set of parameters shown in Table 2. Some analytical details related to Table 2 are in S1 Appendix.

The fifth and sixth rows of Table 1 exemplify another interesting dynamic starting from $I_0 = 1$. $I_t$ becomes larger than $\hat{I}$ (overshoots) and then converges monotonically down to $\hat{I}$; in each case $0 < df(I)/dt|_{I=\hat{I}} < 1$. For the parameters in the seventh row, there is oscillatory convergence to $\hat{I}$ from $I_0 = 1$ ($-1 < df(I)/dI|_{I=\hat{I}} < 0$), while in the last row there is straightforward monotone convergence to $\hat{I}$. The dependence of the dynamics for recursion (11) on the delay Δ and target contact rate $\hat{c}$ is illustrated for Δ = 0, 1, 2 in S6 Fig. The bifurcation diagram for each Δ shows the shift, summarized in Table 2, from convergence to period doubling, chaos, and negative infinity, which occurs for smaller values of $\hat{c}$ as Δ increases.

A continuous-time analog of the discrete-time recursion (11), in the form of a differential equation, substitutes $dI/dt$ for $I_{t+1} - I_t$ in (11). We then solve the resulting delay differential equation numerically using the VODE differential equation integrator in SciPy [42, 43] (source code available at https://github.com/yoavram/SanJose). Using the parameters in Table 2, Figs 1–4 compare the effect of the parameters on the trajectories of the discrete-time and continuous-time SIS model specified in (11). The number of time steps used in the

**Table 2. The effect of the delay, Δ, on dynamics of infecteds*.**

| Δ | Outcome |
|---|---|
| 0 | Monotone convergence to $\hat{I}$ |
| 1 | Damped oscillation to $\hat{I}$ |
| 2 | $\hat{I}$ locally unstable; $I_0 < 72$ bounded oscillation; $I_0 > 73$ unbounded oscillation |
| 3 | $\hat{I}$ locally unstable; collapse ($-\infty$) |
| 4 | $\hat{I}$ locally unstable; collapse ($-\infty$) |

* In all cases, $N = 10,000$, $b_0 = 0.05$, $\gamma = 0.08$, $\hat{c} = 0.0015$, $\alpha = 0.375$, $\hat{I} = 35.718$. $I_0 = 1$ unless stated.

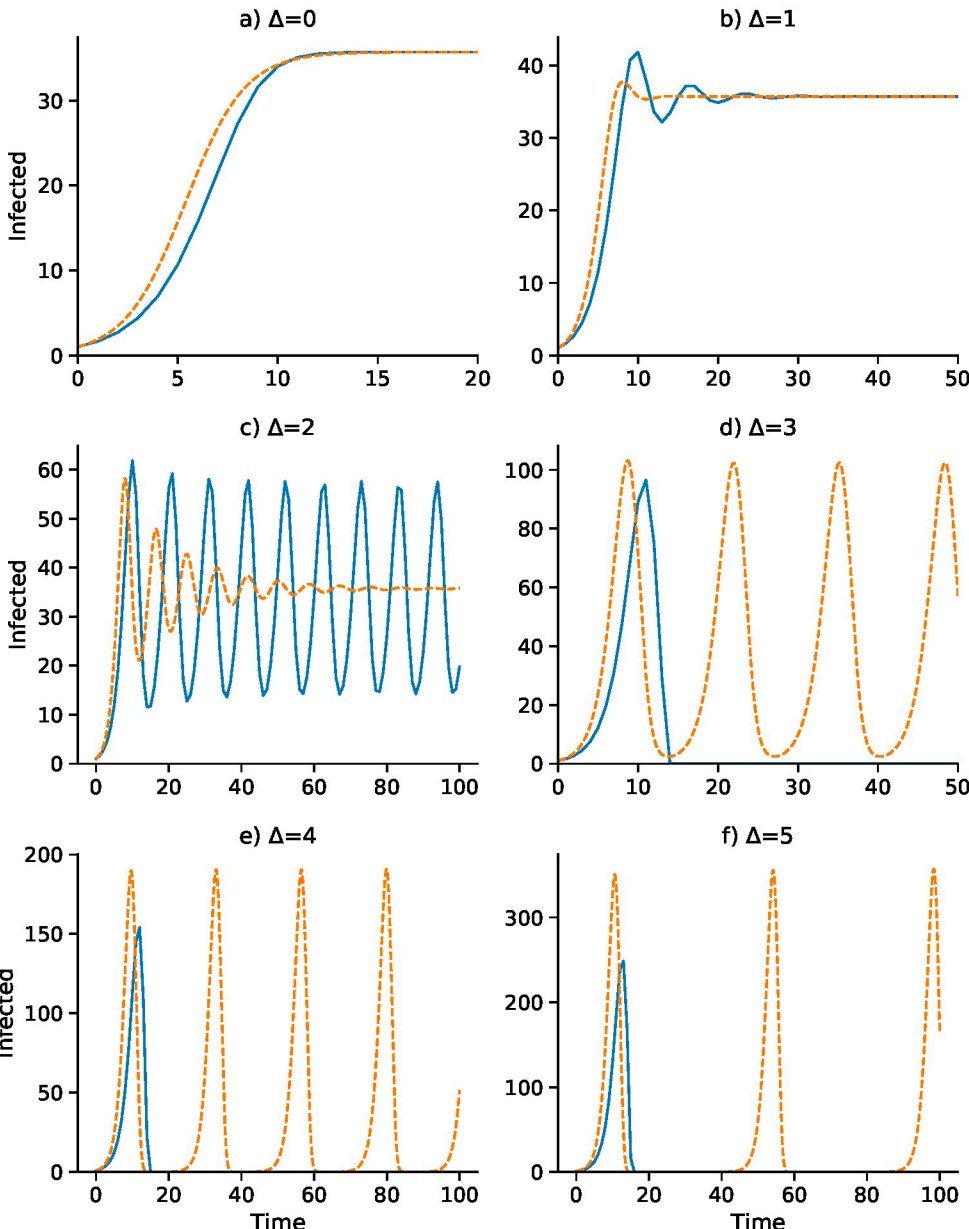

**Fig 1. Discrete-time SIS (blue) and continuous-time SIS (orange) dynamics for delays $\Delta = 0$ to $\Delta = 5$.** $N = 10,000$, $b_0 = 0.05$, $\gamma = 0.08$, $\hat{c} = 0.0015$, $\alpha = 0.375$, and $I_0 = 1$. Here the epidemic equilibrium is $\hat{I} = 35.72$.

computations illustrated in these figures is less than 250 in each case. In Fig 1 the delay ranges from $\Delta = 0$ to $\Delta = 5$, while in Fig 2 the delay is $\Delta = 2$ and Figs 3 and 4 have delay $\Delta = 3$. In the supplementary material S1–S4 Figs, the discrete-time and continuous-time recursions of the SIR model are compared for short and much longer durations.

In Fig 1, with no delay ($\Delta = 0$) and a one-unit delay ($\Delta = 1$), the discrete and continuous dynamics are very similar, both converging to $\hat{I}$. However, with $\Delta = 2$ the differential equation oscillates into $\hat{I}$ while the discrete-time recursion enters a regime of inexact cycling around $\hat{I}$, which appears to be a state of chaos. For $\Delta = 3$ and $\Delta = 4$, the discrete recursion "collapses". In other words, $I_t$ becomes negative and appears to go off to $-\infty$; in Fig 1, this is cut off at $I = 0$.

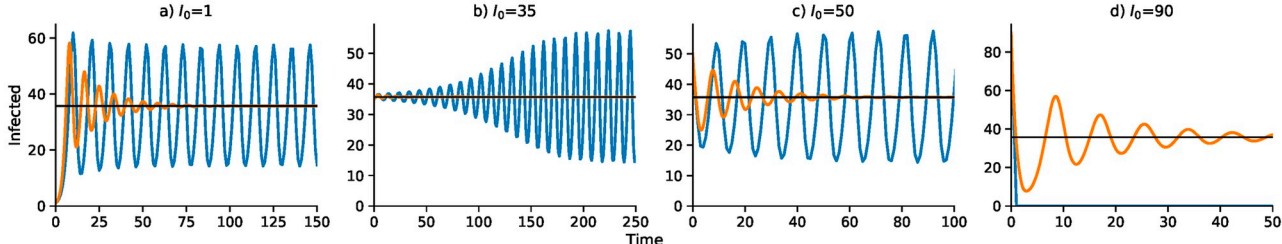

**Fig 2. Effect of initial number of infecteds $I_0$ on the dynamics for delay $\Delta = 2$.** Discrete- and continuous-time results are in blue and orange, respectively. Other parameters as in Fig 1. As in Fig 1, $\hat{I} = 35.72$.

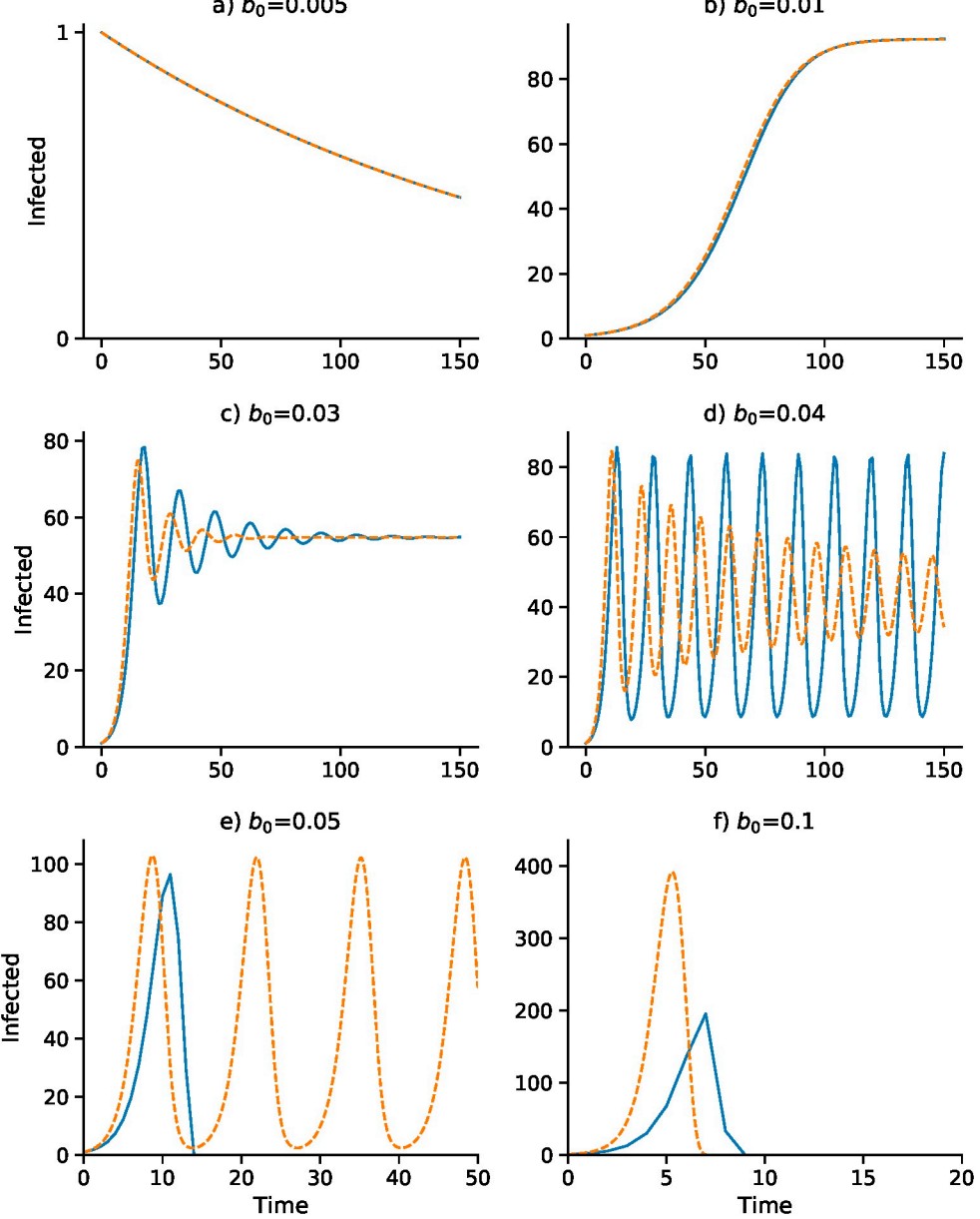

**Fig 3. Effect of baseline contact rate $b_0$ on dynamics with delay $\Delta = 3$.** Other parameters as in Fig 1 with $I_0 = 1$. Discrete- and continuous-time results are in blue and orange, respectively. Note that $\alpha$ changes with $b_0$ as $\alpha = b_0 \alpha_2 / 2\alpha_1$: (A) $\alpha = 0.0375$; (B) $\alpha = 0.075$; (C) $\alpha = 0.225$; (D) $\alpha = 0.3$; (E) $\alpha = 0.375$; (F) $\alpha = 0.75$.

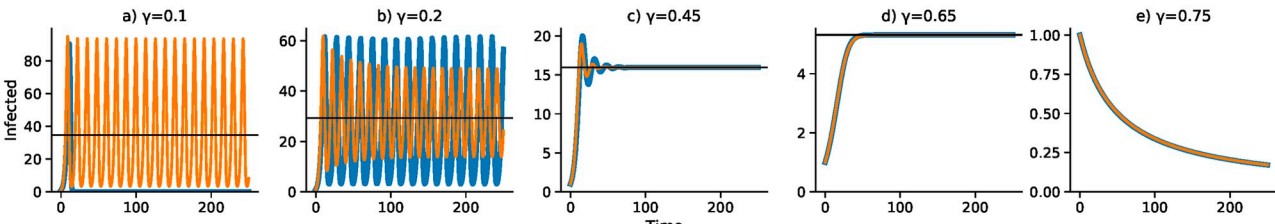

**Fig 4. Effect of removal rate $\gamma$ on dynamics with delay $\Delta = 3$.** Discrete- and continuous-time results are in blue and orange, respectively. Other parameters as in Fig 1 with $I_0 = 1$.

The continuous version, however, in these cases enters a stable cycle around $\hat{I}$. It is important to note that in Fig 1 for each panel the initial frequency was $I_0 = 1$ infected individual. For $\Delta = 2$, for example, with an initial value of $I_0$ higher than about 73, instead of the inexact cycle, which is approached for smaller values of $I_0$, the discrete recursion goes off and becomes negatively unbounded. This dependence of the dynamics on $I_0$ is illustrated for $\Delta = 2$ in Fig 1, where the continuous-time version of the SIS model (11) oscillates into $\hat{I}$. Two expanded views of the inexact cycling seen for $I_0 = 1$ in Fig 1 are presented in S7 Fig.

Figs 3 and 4 focus on a delay of $\Delta = 3$ and show the dependence of the discrete- and continuous-time dynamics on parameters $b_0$ and $\gamma$, respectively. For $b_0$ increasing from 0.005 to 0.05 the pattern of trajectories from $I_0 = 1$ is remarkably similar to that for $\gamma$ decreasing from 0.75 to 0.1. First, both converge to $\tilde{I} = 0$, then both converge to $\hat{I}$, then there is stable oscillation into $\hat{I}$. For $b_0 = 0.04$ and $\gamma = 0.2$, however, the continuous trajectory enters a stable cycle while the discrete trajectory cycles inexactly around $\hat{I}$. For higher values of $I_0$, however, the discrete-time trajectory may become unbounded. Finally, for $b_0 = 0.05$ and $\gamma = 0.75$, the discrete-time trajectory goes to $-\infty$, but is shown stopped at 0, while the continuous case develops a stable cycle.

The discrete- and continuous-time trajectories for the SIR model (13–15) were studied with the same parameters as used in Figs 1–4. Each computation is presented twice: first, for the same length of time as the SIS discrete- and continuous-time in Figs 1–4, and second, for up to 5,000 time units. The trajectories are shown in the Supplementary material, where S1–S4 Figs show short and longer run times. For the longer run times, as expected, in both discrete-time and continuous-time versions of the SIR model, there are eventually no infecteds. Comparing the short-run and long-run figures, the former are not good predictors of the latter in the SIR setting. The short-run behavior of the discrete-time model usually involves a great deal of cycling, which is difficult to see on the longer time scales. S8 Fig compares the SIR and SIS dynamics for the model in Fig 2A with $I_0 = 1$ (see also S7 Fig), with panels A and B illustrating the short term and panels C and D the longer term dynamics. Panels A and B appear to show convergence to $\hat{I}$, but in panels C an D, after about 500 time units, both discrete- and continuous-time versions show the number of infected declining to zero.

It is worth noting that if the total population size of $N$ decreases over time, for example, if we take $N(t) = N\exp(-zt)$, with $z = 50b_0\hat{c}\gamma$, then the short-term dynamics of the SIS model in (11) begins to closely resemble the SIR version. This is illustrated in S9 Fig, where $b_0, \hat{c}, \gamma$ are, as in S8 Fig, the same as in Fig 2A. With $N$ decreasing to zero, both $S$ and $I$ will approach zero in the SIS model, which explain its apparent similarity to the SIR model.

## Discussion

This simple epidemic model with adaptive social contact produces two possible equilibria, one with zero infecteds, where the disease is eradicated, and one between zero and $N$, the

population size, where the disease is endemic. These equilibria are locally stable under different conditions. Dynamics produced by this model are complex and subject to regime shifts across thresholds in the initial conditions and parameter settings. These dynamics include damped oscillation to the equilibrium, periodic oscillation, chaotic oscillation, and regression to positive or negative infinity. Our stability analysis is carried out in the neighborhood of the equilibria. Although global asymptotic stability analysis of some epidemic models has been possible [29, 40, 41], the inclusion of the delay $\Delta$ seems to make global analysis extremely difficult in general [29].

Our model makes a number of simplifying assumptions. We assume that all individuals in the population will respond in the same fashion to government policy and that governments or individuals choose a uniform contact rate according to an optimized utility function, which is homogeneous across all individuals in the population. This contact rate will, in practice, vary across the population according to a variety of drivers including, but not limited to, disease state, cultural and religious practices, political affiliation, housing density, occupation, risk tolerance, and age. Finally, we assume that the utility function is symmetric around the optimal number of contacts so that increasing or decreasing contacts above or below the target contact rate, respectively, yield the same reduction in utility. These assumptions allowed us to create the simplest possible model that includes adaptive behavior trade-offs and time delay.

Convergence to an endemic equilibrium when economic and public health trade-offs are included in an epidemic model is consistent with both theory [22] and other models [33]. Our results show certain parameter sets can lead to limit-cycle dynamics, consistent with other behavior change models [23, 44] and negative feedback mechanisms with time delays [45, 46]. This is because the system is reacting to conditions that were true in the past, but not necessarily true in the present. The time scale and the meaning of the delay, $\Delta$, can influence the qualitative dynamics of the epidemic and, under certain conditions, can lead to a stable cyclic epidemic even in the continuous-time version of our model. We note that these distinct dynamical trajectories as seen in our computational experiments come from a purely deterministic recursion. This means that oscillations and even erratic, near-chaotic dynamics and collapse in an epidemic may not necessarily be due to seasonality, complex agent-based interactions, changing or stochastic parameter values, demographic change, host immunity, or socio-cultural idiosyncrasies. In our discrete-time model, there is the added complexity that the non-zero equilibrium may be locally stable but not attained from a wide range of initial conditions, including the most natural one, namely a single infected individual.

This dynamical behavior in number of infecteds can result from mathematical properties of a simple deterministic system with homogeneous endogenous behavior change, similar to complex population dynamics of biological organisms [47]. The mathematical consistency with population dynamics suggests a parallel in ecology, that the indifference point for human behavior functions in a similar way to a carrying capacity in ecology, below which a population will tend to grow and above which a population will tend to shrink. For example, the Ricker Equation [48], commonly used in population dynamics to describe the growth of fish populations, exhibits similar complex dynamics and qualitative state thresholds. These ecological models are typically structured mathematically in discrete time, while continuous time models are more commonly used in modeling epidemics. There is no a priori reason to prefer the continuous time framework over that in discrete time. It is not clear which strategic approach is more realistic as transmission from an infected to a susceptible individual may happen at anytime, but epidemiologists do tend to frame their thinking in discrete time-steps of days and weeks.

Observed epidemic curves of many transient disease outbreaks typically inflect and go extinct, as opposed to this model that may oscillate perpetually or converge monotonically or

cyclically to an endemic disease equilibrium. Including institutional and public efforts that are further incentivized to eradicate, rather than to optimize short-term utility trade-offs, would alter the dynamics to look more like real-world epidemic curves. Beyond infectious diseases that remain endemic to society, outbreaks may also flare up once or multiple times, such as the double-peaked outbreaks of SARS in three countries in 2003 [49], and surges in fluctuations in COVID-19 cases globally in 2020 [50]. There may be many causes for such double-peaked outbreaks, one of which may be a lapse in behavior change after the epidemic begins to die down due to decreasing incentives [11], as represented in our simple theoretical model. This is consistent with findings that voluntary vaccination programs suffer from decreasing incentives to participate as prevalence decreases [51, 52]. A recent analysis [53] that incorporated epidemic-like transmission of sentiment opposed to vaccination against an infection found that the transient dynamics of the anti-vaccine sentiment could induce complex dynamics of the disease epidemic. However, this analysis did not incorporate a time delay in the manifestation of the anti-vaccine sentiment. The relation between the spread of the sentiment and of the infection is, therefore, somewhat different from that seen here between an adaptive contact rate and the epidemic dynamics.

One of the responsibilities of infectious disease modelers is to predict and project forward what epidemics will do in the future in order to better assist in the proper and strategic allocation of preventative resources. However, there are limits to the power and precision of such modeling. In our model, allowing for adaptive behavior change leads to a system that is qualitatively sensitive to small differences in values of key parameters. These parameters are very hard to measure precisely; they change depending on the disease system and context and their inference is generally subject to large errors. Further, we don't know how policy-makers weight the economic trade-offs against the public health priorities (i.e., the ratio between $\alpha_1$ and $\alpha_2$ in our model) to arrive at new policy recommendations. Geographic and/or cultural variation in our parameter $c_t^*$ (and concomitant variation in the delay $\Delta$) are likely to affect how epidemic dynamics are affected by such trade-offs.

In our model, complex dynamic regimes occur more often when there is a time delay. If behavior change arises from fear and fear is triggered by high local mortality and high local prevalence, such delays are biologically inherent because death and incubation periods are lagging epidemiological indicators. Lags, whether social, environmental, or biological, mean that people can respond inappropriately to an unfolding epidemic crisis, but they also mean that people can abandon protective behaviors prematurely as conditions improve. Developing approaches to reduce lags or to incentivize protective behavior throughout the duration of any lag introduced by the natural history of the infection (or otherwise) should be a priority in applied research. Policy-makers should also consider the benefit of the long-term utility of early-stage overreaction to outbreaks and consider overriding short-term incentives. In light of the COVID-19 crisis, understanding endogenous delayed behavior change and economic incentives is of crucial importance to outbreak response and epidemic management. We anticipate further developments along these lines that could incorporate long incubation periods and other delays, recognition of asymptomatic transmission, influential heterogeneous drivers, and meta-population dynamics of simultaneous, connected epidemics.

## Supporting information

**S1 Appendix. Local stability of the endemic equilibrium $\hat{I}$.** Conditions are given for various values of the delay time $\Delta$ and the parameters in Table 2.
(PDF)

**S1 Fig. Discrete-time (blue) and continuous-time (orange) versions of the SIR model Eqs (13)–(15) with different values of $\Delta$.** Parameters are the same as in Fig 1. Panels A–F represent shorter times and G–L longer times. For $\Delta = 3, 4, 5$, the discrete-time trajectories are stopped at $I = 0$, as they go off to $-\infty$. The continuous-time cases all converge to zero infecteds.
(TIF)

**S2 Fig. SIR version of the SIS model in Fig 2 with $\Delta = 2$ and different values for $I_0$.** Discrete-time (blue) and continuous-time (orange) trajectories are similar to the SIS graphs. Parameters as in Fig 2. Panels A–D represent shorter time and E–H longer times.
(TIF)

**S3 Fig. SIR version of the SIS model in Fig 3 with different values of $b_0$.** Discrete-time (blue) and continuous-time (orange) trajectories are similar to the SIS graphs in Fig 3. Parameters as in Fig 3. Panels A–F represent shorter times and G–L longer times.
(TIF)

**S4 Fig. Effect of removal rate $\gamma$ on discrete-time (blue) and continuous-time (orange) versions of the SIR model.** Note the compression of the cycles seen in Fig 4 and the earlier decline to zero infecteds. Panels A–E represent shorter times and F–J longer times. Parameters as in Fig 4.
(TIF)

**S5 Fig. Bifurcation diagram with varying $\hat{c}$ as in Table 1 on the $x$-axis and its corresponding reproduction number $R_0$.** The dotted horizontal line delineates the total population size ($N = 250$). Dynamics exhibit convergence to the endemic equilibrium (including monotonic, overshooting, and damped oscillation) and period doubling to chaos, followed by passage to negative infinity.
(TIF)

**S6 Fig. Bifurcation diagrams of time delay $\Delta = 0, 1, 2$ as in Table 2 with varying target contact rate $\hat{c}$.** Dynamics progress from convergence to chaos to negative infinity. As $\Delta$ increases, transitions between dynamic regimes begin at smaller values of $\hat{c}$.
(TIF)

**S7 Fig. Dynamics with delay $\Delta = 2$ and initial number of infecteds $I_0 = 1$ in the SIS model (same as Fig 2A).** (**A**): Return map showing more than one $I_{t+1}$ value for each value of $I_t$. (**B**): Comparing the "elliptical" dynamics in part (A) with continuous-time damped oscillation (orange) to equilibrium $\hat{I} = 35.72$. Other parameters as in Fig 2. This figure is the same as Fig 2A.
(TIF)

**S8 Fig. SIR versions of discrete-time (blue) and continuous-time (orange) versions of the SIS model in Fig 2A.** Note the apparent approach to $\hat{I}$ in panels A and B. Both discrete-time and continuous-time trajectories eventually approach $R = N$ for longer times as in panel C.
(TIF)

**S9 Fig. SIS model (recursion (11)) with $N$ decreasing over time.** This uses the same parameters as in Fig 2 but sets $N = N(t) = \exp(-zt)$, where $z = 50\gamma b_0 \hat{c}$ with $\gamma = 0.08$, $b_0 = 0.05$, $\hat{c} = 0.0015$. Note the similarity to S8 Fig, panels A and B.
(TIF)

## Acknowledgments

The authors thank Kaleda Krebs Denton and W. Brian Arthur for helpful comments on an earlier draft of the manuscript.

## Author Contributions

**Formal analysis:** Ronan F. Arthur, Yoav Ram, Marcus W. Feldman.

**Funding acquisition:** Ronan F. Arthur, Yoav Ram, Marcus W. Feldman.

**Investigation:** Ronan F. Arthur, Matthew H. Bonds, Yoav Ram, Marcus W. Feldman.

**Project administration:** James H. Jones, Marcus W. Feldman.

**Supervision:** James H. Jones, Marcus W. Feldman.

**Visualization:** Ronan F. Arthur, Yoav Ram.

**Writing – original draft:** Ronan F. Arthur, Marcus W. Feldman.

**Writing – review & editing:** Ronan F. Arthur, James H. Jones, Yoav Ram, Marcus W. Feldman.

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
