## [Decision Letter · Decision Letter 0]

3 Oct 2020

Dear Dr. Feldman,

Thank you very much for submitting your manuscript "Adaptive social contact rates induce complex dynamics during epidemics" for consideration at PLOS Computational Biology.

As with all papers reviewed by the journal, your manuscript was reviewed by members of the editorial board and by several independent reviewers. In light of the reviews (below this email), we would like to invite the resubmission of a significantly-revised version that takes into account the reviewers' comments.

I enjoyed reading the manuscript and particularly the relevance of the idea that behavioral changes elicited by potentially outdated information can result in complex dynamics for the epidemic. Reviewer 1 has made a number of very reasonable comments which can and should be addressed. Please discuss the relevance of SIS vs SIR dynamics more clearly. For coronaviruses it seems that the dynamics lies somewhere between SIS and SIR --- infection induces immunity that: (i) very significantly reduces pathology; (ii) transiently prevents infection and (iii) reduces the extent of transmission following subsequent infections. While including these details is not likely to change the main conclusions of the paper it may be worth mentioning in the discussion section.

We cannot make any decision about publication until we have seen the revised manuscript and your response to the reviewers' comments. Your revised manuscript is also likely to be sent to reviewers for further evaluation.

Sincerely,

Rustom Antia

Associate Editor

PLOS Computational Biology

Stefano Allesina

Deputy Editor

PLOS Computational Biology

I enjoyed reading the manuscript and particularly the relevance of the idea that behavioral changes elicited by potentially outdated information can result in complex dynamics for the epidemic. Reviewer 1 has made a number of very reasonable comments which can and should be addressed. Please discuss the relevance of SIS vs SIR dynamics more clearly. For coronaviruses it seems that the dynamics lies somewhere between SIS and SIR --- infection induces immunity that: (i) very significantly reduces pathology; (ii) transiently prevents infection and (iii) reduces the extent of transmission following subsequent infections. While including these details is not likely to change the main conclusions of the paper it may be worth mentioning in the discussion section.

Reviewer's Responses to Questions

**Comments to the Authors:**

Reviewer #1: Epidemics have huge public health and economic effects, and governments have the challenge to balance the costs and benefits in considering different interventions. While lockdown restrictions may curtail spread, the economic costs of such policies may be great. In this paper, the authors highlight the importance of understanding how human behaviour responds to an epidemic and thereby changes patterns of transmission. These dynamics need to be considered in establishing government responses to epidemics. The authors develop and analyse an epidemiological model (both continuous and discrete time, and SIS and SIR) where social contact rate is modelled as a utility function with delayed information. Individuals receive outdated information and thus they are responding to 'old' information. This lag in the system causes complex dynamics. They find that the dynamics are highly sensitive to details such as initial conditions and the nature of the behavioural response to infection dynamics, and therefore the time-delays. Thus, it is inherently difficult to make accurate predictions from models without further studying and measuring these effects.

These are significant messages and the analysis demonstrates the principles clearly. However, the manuscript struggles with the tension between realism and tractability. Understanding the current pandemic is an important goal but perhaps slightly at odds with the central theme here.

Specific comments and queries follow.

1. From the abstract the authors frame the problem in terms of COVID-19. However, this link to COVID-19 does not seem essential to the main messages in that parameters are not calibrated to any particular data (observational or quantitative). Introducing this link early in the article suggests a direct and focused application of the model to COVID-19. But this doesn't appear to be the aim of the study.

One way to address this issue is to re-frame the early part of the manuscript around the more general problems of time-delays, behavioural change and potential unpredictable epidemic dynamics. And slightly reduce emphasise on COVID-19.

On a related note, the authors' text in lines 326-330 regarding Holling's heuristic distinction in ecology effectively signal the authors' intentions. This would be useful to state earlier in the text as well (or instead). This may help stave off criticisms of parameter choice etc. that are not data-driven.

And again along the same lines, the authors suggest that this work may have “a useful implication for policy” (line 360). It is not completely clear what this is. If prediction is “perhaps impossible” (line 376), then what might be next and what does this mean for public health? Some discussion on whether or not this is useful would benefit the paper.

2. Figures 1-4 are all based on the SIS model. SIS is unlikely to be a good description of COVID-19 if there is immunity (and hopefully there is and an effective safe vaccine will be found). It's clear this study is not just about COVID, but is there a reason to focus the analysis on SIS dynamics, which might be very different to COVID dynamics? Is this because SIS is more tractable?

In the standard SIR model the epidemic ends as the infectious cases encounter too many recovered and not enough susceptible people, assuming a constant contact rate. It would be interesting to explore in your model whether the oscillations alter these dynamics and sustain the epidemic where it otherwise would have ended.

3. Is there a basic reproduction number or equivalent in this system – an R_0 or R_t? Is there a threshold effect? In Fig 3 for example, it appears as if crossing a threshold causes a shift in behaviour. This could perhaps be addressed with additional text.

4. How herd immunity is handled also reflects a mismatch between the analysis and the goal to understand COVID. In line 175 you say “... we do expect that from any initial frequency I_0 of infected all N individuals will eventually be in the R category” and in line 294 “For the longer run times, as expected, both discrete-time and continuous-time versions of the SIR model eventually converge to R=N with no infected”. It is not clear why this should be so. In standard SIR models R does not converge to N unless R_0 goes to infinity. I(t) does go to zero, but not all susceptibles are infected (S(t) does not go to 0), so R(t) should not go to N. Explain/clarify or re-examine. It would be helpful to plot the removed class R(t). Presumably it approaches herd immunity levels. It would be horrific if we really expected everyone to be ultimately infected with SARS-CoV2!

5. The differences in trajectories between discrete and continuous time suggest that the formulation of the model makes a big difference. Does this mean that the complexity of dynamics is partly an artefact of how the model is formulated? Which is more realistic or better: discrete or continuous? How much is mathematical curiosity (i.e. chaos in logistic map model) and how much is actually of interest in a real epidemic context?

6. The model is clearly sensitive to initial conditions and parameters. How do you know that the long-term behaviour is accurately determined by the numerical analysis? Is it possible that small numerical errors accumulate and grow, rather than diminish over time?

7. Since the model is sensitive to these differences in initial conditions and parameters, could the authors synthesise the interpretation of these results in terms of the effect of different parameter choices? What do the parameter choices mean in terms of COVID-19, or other epidemics? E.g. do you only get monotone convergence when the target contact rate is very low? While the conditions are given, some interpretation of 'ballpark realism' may be useful.

Minor technical comments:

1. In line 193 and continued throughout – “illegitimate” is a funny word choice here relating to I*. Perhaps “physically unreal/impossible” or “not feasible” could be used instead?

2. Fig 4 curves are thick and blobby. Could these be made slightly thinner?

3. Line 136: "The second term, \\alpha_1 (c − \\hat{c})^2 , is a concave function" - do you mean convex? With the negative sign it is concave.

4. There are lot of interesting results with different behaviour as listed in Table 1. However, the presentation is quite hard to follow. Since these are numerical examples (the placing of these under “analytical results” doesn't seem quite right either), could these be related to the Figures in a concise/useful way to help the reader understand the information?

5. Supplementary Figures in general: it would be good (if the journal requirements permit) to combine all the figures into a single document which includes figure legends. The alternative is quite awkward to look through.

6. FigS1 looks the same as Fig1. Is it supposed to be?

7. In the long-time Supplementary Figures it is sometimes hard to tell what's going on. E.g. sometimes hard to tell where the blue trajectories go. Is there any other way of making comparison? You could try log scale for time, though admittedly that would be a bit unconventional.

8. Could the authors add a brief comment/prediction about what they expect to happen if there was asynchrony in the delay term? Presumably not everyone follows the same behaviour and may be slower to adopt restrictions.

Reviewer #2: equation 18: full stop missing

intro paragraph at line 80 - drop in some citations for the covid 19 delays.

lin 121: definition of c*_t as the optimal value apear twice in that long sentence

equation 6: your aproximation is equivalent to low prevalence. suggest stating that here, rather than later on.

line 145: in that paragraph you focus on optimal number of contacts. For individuals at high risk of severe disease, the chance of negative outcomes given N contacts is likely higher than for a low risk individual. Perhaps expand on this limitation a bit more at line 322? Because the assuming a high risk individual has far fewer contacts than a low risk individual would likely affect estimates of the burden on healthcare systems. eg UK covid policy involved high risk individuals 'shielding', so having different rules to the rest of the population. So whilst a better understanding of behavioural dynamics could improve model predictions, it is important to ensure they allow for hetrogeneity.

line 156: you introduce c* having previously introduced c*_t. You then redefine c* as c*_t and then go on to mention both. I assume you require only one of these, c*_t?

line 198: you mention the importance of global stability analysis then immediatly follow with local instability and stability only, therefore the statement seems out of place. Please remove or clarify.

justify why looking at an SIS model, and then and SIR model. I think a motivating sentence to lnk together the introduction to SIS, then SIS to SIR would be helpful.

Can you justify your choice of modelled delays? In the introduction you mention a delay of 14 days (line 87) for public health to understand the effects of interventions, however the results section has a maximum delay of 5 days.

I think the first paragraph or two of the discussion would benefit from more real-world disucssion of what the results mean. Currently this comes later in the discussion, after limitations.

In the manuscript you assume disease prevalence is low and make a first order approximation. Perhaps you have an opportunity to link to covid in the discussion as many countries have interventions in place so covid is a disease that is remaining at low prevalence.

line 373: "COVID-19 models have often proved wrong by orders of magnitude because they lack the means to account for adaptive response."

Unsupposted statement.

I am also unsure that it should remain - I am unsure if the models that are feeding in to policymakers are all published yet, so do citations exist to back this statement as it stands?

You may want to rewrite this paragraph a bit. I think you have scope to say that existing models are perhaps not fully informed about behaviour so have to make simplifying assumptions, or that certain relevant intervention compliance questions are not able to be answered by existing models.

Similarly, in the abstract, "Models have proved inaccurate because behavioral

response patterns are either not factored in or are hard to predict." This again very much depends on what questino the models are trying to answer. If they are well-informed from, say, hospital data then they might be very good at predicting bed demand, even without a mechanistic model for understanding behaviours in the community.

**Have all data underlying the figures and results presented in the manuscript been provided?**

Reviewer #1: Yes

Reviewer #2: None

PLOS authors have the option to publish the peer review history of their article (what does this mean?). If published, this will include your full peer review and any attached files.

Reviewer #1: No

Reviewer #2: **Yes: **Caroline E Walters
---

## [Decision Letter · Decision Letter 1]

16 Dec 2020

Dear Dr. Feldman,

We are pleased to inform you that your manuscript 'Adaptive social contact rates induce complex dynamics during epidemics' has been provisionally accepted for publication in PLOS Computational Biology.

Best regards,

Rustom Antia

Associate Editor

PLOS Computational Biology

Stefano Allesina

Deputy Editor

PLOS Computational Biology

Reviewer's Responses to Questions

**Comments to the Authors:**

Reviewer #1: The authors have considered and addressed the suggestions and queries raised in the first review. For example there is a better balance between the mathematical/computational analysis and the epidemiological implications, and the discussion of the SIR model is updated and an R0-based threshold introduced. By reframing the manuscript as a theoretical investigation rather than a more direct application to COVID-19, the wider benefits of the article are made clearer.

The minor issues have been addressed.

One remaining problem: a repeated phrase in lines 109-112. The interpretation of c_t^* is given twice.

Reviewer #2: Thanks to the authors for all their work on the manuscript. Removing the focus of covid-19, framing as a more theoretical piece, has really improved the manuscript in my opinion. Lot easier to read, both introduction and discussion provide clear justification for the work and how it contribution the wider literature. If you decide to do another paper with behavioural heterogeneity then I look forward to reading it.

minor:

line 194 "legitimate" still used.

line 182: “Assessing global asymptotic stability in epidemic models is an important task of mathematical epidemiology”

I really think you don’t need this sentence right here, because you do not do global stability analysis. I understand why, that’s fine, I’d just not have this sentence in the results section. Just move it to the discussion. Lead with ‘this is what we did’ or just lead with “The three equilibria of the SIS recursion…”

**Have all data underlying the figures and results presented in the manuscript been provided?**

Reviewer #1: Yes

Reviewer #2: Yes

PLOS authors have the option to publish the peer review history of their article (what does this mean?). If published, this will include your full peer review and any attached files.

Reviewer #1: No

Reviewer #2: **Yes: **Caroline E Walters

---

## [Editor Report · Acceptance letter]

25 Jan 2021

PCOMPBIOL-D-20-01217R1 

Adaptive social contact rates induce complex dynamics during epidemics

Dear Dr Feldman,

I am pleased to inform you that your manuscript has been formally accepted for publication in PLOS Computational Biology. Your manuscript is now with our production department and you will be notified of the publication date in due course.

With kind regards,

Alice Ellingham
